# Simulating Mass in Virtual Reality using Physically-Based Hand-Object Interactions with Vibration Feedback

Hooman Khosravi*          Katayoon Etemad†          Faramarz F. Samavati ‡

University of Calgary

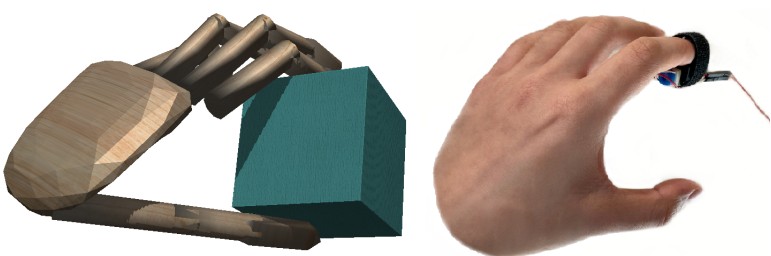

Figure 1: Physics-based interactions with virtual objects using a co-located virtual hand (the left figure) are augmented using vibrational feedback proportional to objects' mass and acceleration (the right figure).

## ABSTRACT

Providing the sense of mass for virtual objects using un-grounded haptic interfaces has proven to be a complicated task in virtual reality. This paper proposes using a physically-based virtual hand and a complementary vibrotactile effect on the index fingertip to give the sensation of mass to objects in virtual reality. The vibrotactile feedback is proportional to the balanced forces acting on the virtual object and is modulated based on the object's velocity. For evaluating this method, we set an experiment in a virtual environment where participants wear a VR headset and attempt to pick up and move different virtual objects using a virtual physically-based force-controlled hand while a voice-coil actuator attached to their index fingertip provides the vibrotactile feedback. Our experiments indicate that the virtual hand and our vibration effect give the ability to discriminate and perceive the mass of virtual objects.

**Index Terms:** Human-centered computing—Human computer interaction (HCI)—Interaction devices—Haptic devices; Human-centered computing—Human computer interaction (HCI)—Interaction paradigms—Virtual reality

## 1 INTRODUCTION

Virtual Reality (VR) has significantly revolutionized simulated human experiences. VR enables an immersive virtual experience by simulating and triggering most of our senses as if we are present in another environment. Notably, in VR it is possible to see one's own co-located virtual hands, perceive them as their own real hands and interact with virtual objects [16]. However, virtual objects have no real mass, and the problem is including touch and visual cues that we rely on for mass perception. The physical cues include skin stretch and contact pressure at the fingertips (cutaneous feedback) and proprioceptive feedback from multiple muscles and joints (kinesthetic feedback).

Grounded haptic devices can render the necessary forces for kinesthetic and cutaneous haptic feedback. However, their size,

---
*e-mail:hooman.khosravi1@ucalgary.ca
†e-mail:ketemad@ucalgary.ca
‡e-mail:samavati@ucalgary.ca

weight, and limited workspace restrict free-hand movements, making them less desirable in various VR applications.

Alternatively, ungrounded haptic devices (such as finger-mounted or hand-held devices) can be built more compactly and lighter, making them more convenient to use in a larger workspace. Sensing the mass of a virtual object in every direction needs more complex ungrounded hardware with higher degrees of freedom. However, such devices require multiple actuators and can limit hand and finger movements.

Another approach to overcome the hardware limitations is to use visio-haptic illusions. These methods aim to trick the brain into perceiving the mass by manipulating the objects' visual cues. For example, limiting the virtual object's velocity [1], or scaling its displacement compared to the user's hand [21] are shown to give a sense of mass to the objects. However, these methods are not physically realistic or decrease the co-location between the actual and virtual hands.

In this paper, we introduce a novel mass rendering method that combines a visio-haptic technique with a simple finger-mounted vibration actuator. For the visio-haptic part, we replicate the visual cues that humans perceive during a real-world hand interaction with physical objects. We use a force-controlled physically-based virtual hand in VR to interact with virtual objects, which results in a limit on the heaviness of the objects that the user can pick up and how fast they can accelerate them based on their mass. However, it is difficult to distinguish between light objects using this technique. We complement our visio-haptic method with haptic feedback. The haptic actuator that renders the feedback should be small and compact enough to allow individual fingers to move independently and perform dexterous interactions. Also, we prefer an ungrounded device since it allows a larger workspace. One method to reduce the device's size is to use haptic feedback that is directionally invariant to our sense of touch. If the haptic stimulus's direction is detectable by the sense of touch, we need multiple actuators to render the haptic effect in different directions during a virtual interaction. Therefore we employ an ungrounded, direction invariant haptic effect to complement our physically-based virtual hand. We explore using a mechanical vibration feedback effect to achieve ungrounded mass rendering for virtual objects. In our work, while the virtual object is in the user's grasp, a sinusoid vibration proportional to the object's mass and acceleration is played through an ungrounded voice-coiled actuator at the tip of the user's index finger. An overview of the proposed method is shown in Fig. 1.

When moving two objects with different mass, in addition to the physically-based visio-haptic feedback, users feel proportionally stronger vibration while grasping the heavier object. This vibrotactile feedback gives the user a clue to the net force acting on the virtual object. To make this a direction-invariant haptic feedback, we use frequencies above 100Hz. These frequencies are sensed by Pacini mechanoreceptors, which are not sensitive to the stimuli's directions.

To evaluate the proposed physically-based virtual hand and the vibration feedback, we conducted a user study where participants interact with virtual objects with different masses and perform virtual tasks. Using qualitative and quantitative methods, we show that the physically-based hand gives a sense of mass to virtual objects, and adding the vibration feedback does improve mass perception and discrimination.

The main contribution of this work is the design, development, and evaluation of a novel mass rendering method for virtual objects using physically-based hand-object interactions and vibration feedback.

## 2 RELATED WORKS

In this section, we review relevant literature on simulating the mass of virtual objects during a VR experience. Also, we discuss modes of interaction in VR, including physically realistic grasping and interaction.

Grounded haptic devices are highly sought-after in tool-mediated applications where precision and fidelity are essential such as surgical training [11]. Hand wearable grounded devices have also been developed. HIRO III [10] is an example of a five-fingered grounded haptic interface, with three DoF for each of its haptic fingers and a 6 DoF base capable of providing high precision force feedback to a hand while it is attached to each of the fingertips. The main challenge with grounded devices is their limited workspace size.

Ungrounded haptic devices are attached to the user's body instead of a fixed point in the room, which allows a larger workspace. These devices are either hand-held or attached to the user's fingers, hands, or body. Minamizawa et al. [19] introduce a fingertip mounted ungrounded haptic device called the Gravity Grabber that can create a sense of weight when grabbing virtual objects in specific orientations. Gravity Grabber achieves this using one degree of freedom for shear force feedback and another degree of freedom in the normal direction of the fingertip skin. However, since our skin can detect the direction of skin stretch, this method cannot give a sense of weight to a virtual object in all orientations. Sensing the weight and inertia of a virtual object in all directions requires an ungrounded device with more complex hardware and higher degrees of freedom. Such as the works of Chinello et al. [4], and Prattichizzo et al. [20]. However, such devices are mechanically complicated since they require multiple actuators and limit hand and finger movements. In our method, we use one haptic actuator to render the mass of objects in all directions since we use sinusoidal vibration feedback.

Hand-held ungrounded devices are desirable for simulating interactions with hand-held tools such as a hammer or a baseball bat. However, they limit the movement of fingers and the hand. Zenner in [26] introduced Drag:on a custom VR hand controller with two actuated fans, which can dynamically adjust the controller's aerodynamic properties, therefore changing the sensed inertia of a virtual object. Zenner et al. [25] introduce Shifty, a hand-held VR controller with an internal prismatic joint connected to a weight that shifts the center of mass of the device, resulting in different rotational inertia and resistance as the user interacts with various virtual tools. In the work of Lykke et al. [17], users have two hand controllers to pick up round virtual objects (scooping), and they should keep their hands closer together when the objects are heavier. Our method tracks the user's own hand instead of using a VR controller, which increases the sense of ownership and realism of the virtual hand [16] while not limiting the fingers' and hand's mobility.

Humans can use visual cues to determine the weight of a virtual object. Backstrom [1] gives the sensation of mass to virtual objects in VR by limiting the velocity of a virtual object based on how heavy it is. Such constraints on the object's movements are not physically realistic. Dominjon et al. [9] show that manipulating the control-display ratios of virtual objects can change the perceived mass in virtual environments. In other words, if a virtual object's displacement is proportionally increased compared to the user's actual hand, its mass is perceived as lighter than it is and vice versa. Samad et al. [21] utilize the same technique in VR to change the perceived weight of wooden cubes. However, one downside of changing the control-display ratios is that the offset between the actual and the virtual representation of the object increases as the hand gets further away from the initial contact point. Therefore, bi-manual coordination and interaction could become difficult since the virtual hand's relative position is different from the actual hand's, even if it is not moving. Our approach aims to give a sense of mass to objects by using a physically-based virtual hand that enables realistic interactions with virtual objects and preserves the co-location between the virtual and actual palm when the hands are steady or when their acceleration is not changing.

Interaction is an important part of an immersive virtual experience and increases the user's sense of presence [3] [24]. There are various ways to enable interactions between a virtual hand and virtual objects. In gesture and metaphor-based approaches, the interaction uses specified hand commands. For example, if the virtual hand is in a grasping pose and near a virtual object, that object's orientation follows the virtual hand. Song et al. [23] enables 9 DoF control of a virtual tool using bi-manual gestures. Gesture-based approaches have proven to be robust and effective. However, they are unintuitive and artificial by nature; therefore, they are not suitable for a physically realistic interaction. Another approach is to use physically-based manipulation techniques. For example, Borst and Indugula in [2] propose virtual coupling of the tracked hand to a rigid virtual hand that enables whole hand grasping. In this method, the palm and finger joints of the tracked hand and the virtual hand are connected to the corresponding parts using linear and torsional virtual spring-dampers. Moreover, since the spring damper links work based on applying a limited and proportional amount of force, this method shares the same physical limitations that a realistic interaction has. We modify this method to preserve the co-location between the virtual and actual palms and evaluate it for mass rendering in VR.

Vibration feedback can be used to simulate different touch stimuli. We use sinusoidal vibrations to render the mass of a virtual object. Asymmetrical vibration is another type of vibration feedback that has been used by Choi et al. [5] to simulate weight in VR. These vibrations cause skin-stretch, and the user can detect their direction. Therefore, multiple actuators are required for simulating weight and inertia in all directions. Moreover, the intensity of these asymmetrical vibrations is much stronger (up to 20 g($9.8\,\mathrm{m\,s^{-2}}$)) compare to our vibration feedback (less than 1 g). Kildal [13] uses grain mechanical vibrations to create the illusion of compliance for a rigid box. Sinusoidal vibration feedback has been used in other haptic applications such as simulating a button press on a rigid box [15] and a virtual button in VR [14]. Moreover, Seo et al. [22] simulate a moving cart by adding vibration feedback to a chair and changing the amplitude and frequency of the vibration feedback proportional to the simulated cart's angular velocity.

Mass rendering methods in VR limit the hand and finger movements or engage users in unrealistic interactions. Our physically-based interaction is realistic and preserves the co-location of actual and virtual palms when the hands are under no or constant acceleration, and our vibration feedback works with a single actuator on the fingertip without limiting the hand and finger movements.

## 3 FORCE-CONTROLLED VIRTUAL HAND

One of the goals of this paper is to explore the effect of a physically-based interaction on mass perception and discrimination. There is a weight limit on objects in the real world that we can pick up using our hands. Our grip strength and the force that we can apply to a grasped object are bounded. Therefore, there is a limit to how fast we can accelerate an object based on its mass. In VR, we hypothesize that physically-based interaction between the user's virtual hand and object creates a sense of mass for that object. For this purpose, we track the user's hand, couple it with a 3D model of a hand, and use a physically-based simulation for hand-object interactions. We use a vision-based hand tracking system (Leap Motion hand tracker) to allow the user's hand and fingers to move freely, providing a virtual experience analogous to real-world interaction.

For modeling the hand, we consider one rigid palm and five fingers, each of which has three rigid phalanges. Interaction between VR objects and the force-controlled virtual hand is more realistic than interactions between the tracked hand and VR objects. For example, when grasping an object, the tracked hand can go inside the object, but the virtual hand grasps around the object. Therefore, we only display the force-controlled virtual hand (VR hand). The VR hand must be co-located and coupled with the tracked hand. To achieve this, rather than a purely geometric approach, we modify the physically-based method described by Borst and Indugula in [2]. The physically-based coupling helps us to efficiently prevent unrealistic collisions and interactions between the VR hand and objects. In the physically-based coupling method, we associate one spring-damper to each rigid component of fingers. The spring-dampers apply force to the VR hand's components to match their positions and orientations to the tracked hand's corresponding components. To achieve consistent behavior from the physical simulation, we use a fixed size VR hand. Having a fixed size for the VR hand does not directly influence efficiency in virtual object manipulation tasks, sense of hand ownership, realism, or immersion in VR [16].

The spring-damper coupling applies both force and torque to the virtual part. The force at time $t$, $\vec{F}(t)$, is proportional to $\Delta_{Position}(t)$, the distance between the center of the mass of the two corresponding parts and the torque at time $t$, $\vec{\tau}(t)$, is proportional to $\Delta_{Rotation}(t)$, the difference in their rotation. To prevent the virtual part from overshooting its target position and orientation, the spring-damper applies another force to the virtual object proportional to $\vec{V}(t)$, its linear velocity and torque proportional to $\vec{\omega}(t)$, its angular velocity. That gives:

$$\vec{F}(t) = k'_p \vec{\Delta}_{Position}(t) - k'_d \vec{V}(t), \tag{1}$$

$$\vec{\tau}(t) = k''_p \vec{\Delta}_{Rotation}(t) - k''_d \vec{\omega}(t) \tag{2}$$

where $k'_p$, $k''_p$, $k'_d$ and $k''_d$ are the spring-damper coefficients. These parameters, are set during the preliminary experiments to ensure that the VR hand is responsive and closely and smoothly follows the actual hand and can pick up virtual mass up to 4kg.

If we use a similar spring-damper to couple the palms, when the user holds an object using the VR hand, the distance between the VR hand and the actual hand increases until the spring-dampers' forces equal the weight of the VR hand and the object that it is holding. This causes a discrepancy between the visual and the proprioceptive sense. To solve this problem, we introduce an additional term in the spring-damper for the palms:

$$\vec{F_{Palm}}(t) = k'_p \vec{\Delta}_{Position}(t) - k'_d \vec{V}(t) + k'_i \sum_{j=0}^{t} \vec{\Delta}_{Position}(j), \tag{3}$$

$$\vec{\tau_{Palm}}(t) = k''_p \vec{\Delta}_{Rotation}(t) - k''_d \vec{\omega}(t) + k''_i \sum_{j=0}^{t} \vec{\Delta}_{Rotation}(j) \tag{4}$$

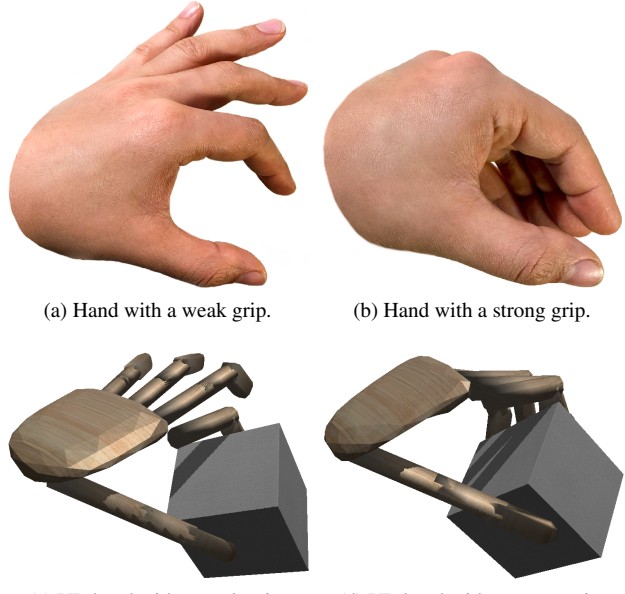

| | |
|---|---|
| (a) Hand with a weak grip. | (b) Hand with a strong grip. |
| (c) VR hand with a weak grip. | (d) VR hand with a strong grip. |

Figure 2: A weak and a strong virtual grip and the corresponding actual hands.

where $k'_i$ and $k''_i$ are spring-damper coefficients. The added summation term applies force and torque proportional to the accumulation of $\vec{\Delta}_{Position}(t)$ and $\vec{\Delta}_{Rotation}(t)$ over time. Therefore, when the user holds an object, $\vec{F_{Palm}}(t)$ and $\tau_{\vec{Palm}}(t)$ increase until the virtual palm's orientation and position match the tracked hand palm in the steady-state. $k'_i$ and $k''_i$ are set during the preliminary experiments so that position and orientation of the coupled palms quickly match when the hand is not accelerating. Also, $k'_p$, $k''_p$, $k'_d$ and $k''_d$ are set independently for the palm compared to the phalanges since it has different physical properties.

Using a force-controlled virtual hand should give a sense of mass perception and allow mass discrimination between virtual objects. However, we suspect that this claim is stronger in some scenarios and weaker in others. While grasping and moving a light object, the spring-damper forces counteract the force of gravity and inertia on the object. Therefore, using our virtual hand, if a user grasps an object with a low virtual mass, they can easily pick it up and quickly move it around the workspace with high acceleration without it coming out of their grip. However, for a heavier object, the user can still pick it up, but they have to increase their effort, such as using more fingers for grasping or closing their grip further so spring-dampers would apply more force on the object (Fig. 2). Also, it is not possible to accelerate it as fast as lighter objects since the inertial forces are higher and can overcome the spring-dampers in the virtual hand and open the virtual grasp. Depending on the spring dampers' coefficients, after a certain point in mass, it would be really difficult or eventually impossible for the user to move or pick up the object. We hypothesize that the limit on how fast the user can accelerate the virtual object in hand and how challenging it is to pick it up gives the user a sense of the virtual object's mass and enable them to discriminate two objects based on their mass. However, using this technique, it is hard to perceive the difference in mass between two light objects (<1kg) since it would be almost effortless to pick both of them up off the ground and move them quickly without dropping them. To overcome this problem, we introduce a vibration feedback effect to complement our VR hand.

## 4 VIBROTACTILE FEEDBACK

In day-to-day physical interactions with real-world objects, we can feel the object's mass and compare it to other heavier or lighter objects through our sense of touch. Virtual experiences that do not provide haptic feedback lack realism compared to real-world experiences. One of the modalities of haptic feedback is vibrotactile feedback in the form of mechanical waves or vibrations.

Our goal is to complement the VR hand in giving the user a perception of an object's mass by communicating the net force they apply to the object. To achieve this without limiting the hand and finger movements, we use one actuator to render our haptic feedback. We use sinusoidal vibration feedback with a frequency range between 100Hz and 150Hz, making it perceivable only by the Pacini mechanoreceptors in the fingertip skin. The Pacini mechanoreceptors cannot detect the direction of the mechanical waves; therefore, only one actuator is sufficient to render our haptic feedback in all directions.

We strap a VCA (voice-coil actuator) to the fingertip of the index finger. We chose the index finger because it has a critical role in picking up objects with a pinch grasp. Other fingers, such as the thumb and the middle finger, can have an important role in grasping as well; However, attaching voice-coil actuators to multiple fingers limits the relative movement of fingertips and manual dexterity.

While a user grasps an object, we render the vibration feedback $O(t)$ with frequency $O(t)_F$. The amplitude of $O(t)$ is proportional to the object's mass $M$ and acceleration $A(t)$. This results:

$$O(t) = \alpha MA(t)sin(2\pi t O(t)_F), \tag{5}$$

where $\alpha$ is a scaling constant to control the range for the vibration energy perceived by the user. The vibration feedback should be only strong enough so that users can perceive the vibration when slowly moving the lightest weight in the scene. The value of $\alpha$ also depends on the hardware components of the haptic chain, such as the signal amplifier and the haptic actuator. For our setup, we set the $\alpha$ value in a way that, if the user accelerates a 1kg object at 1 g, the measured vibration at the fingertip is on average 0.32g, which allows users to perceive the vibration feedback when slowly moving the lightest weight (0.25kg) in our experiments. The frequency of the output signal $O(t)_F$ dynamically changes from 100Hz to 150Hz based on the velocity of the virtual object $V(t)$, that gives:

$$O(t)_F = max(150, 100\frac{|V(t)| + 2}{2}), \tag{6}$$

where at speeds near zero, the signal's frequency is 100Hz, and as the speed increases to about one m/s, it goes up to 150Hz. To ensure a smooth vibration signal, we apply a second-order Butterworth lowpass filter to $V(t)$ and $A(t)$. The filter has a sample rate of 1000Hz, and the corner frequency is 20 Hz(-3db amplification at 20Hz).

We set the signal's amplitude proportional to $MA(t)$ which, according to Newton's second law of motion, represents the net force acting on the virtual object. In our method, we ignore balanced or counteracted forces acting on an object since the counteracted forces from grasping can be similar between a light and a heavy object. As an example, we can grip a light object just as hard as a heavier one.

During a virtual experience, the voice-coil actuator is always strapped to the user's index fingertip. However, the vibration feedback renders only when the user's virtual hand grasps a virtual object and not during their free-hand motions in the scene. To detect if the user is grasping a virtual object, we check whether the virtual object is off the ground and touching the virtual hand's palm and the distal joint of the thumb, index, or middle finger. If grasping is detected, the vibration feedback is rendered for the user through the voice coil actuator.

Whenever the system detects that the user is no longer grasping a virtual object, the vibration feedback rendering stops. However,

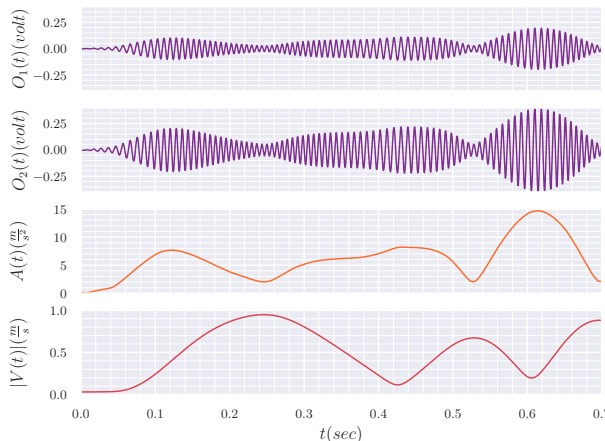

Figure 3: The output voltage of the vibration feedback for two virtual objects with mass values 0.5kg, $O_1(t)$, and 1kg, $O_2(t)$, during an arbitrary shaking movement with acceleration, $A(t)$, and velocity $V(t)$.

in a physical simulation, even when the user is grasping the object, the hand parts may momentarily lose contact with the virtual object for a few cycles, and this might cause on/off pulses in our vibration feedback. To avoid these impulse noises in our signal, we stop the vibration feedback after no grasping is detected for ten milliseconds.

When the user picks two virtual objects with different mass values and moves them around the scene with the same motion, the vibration effect is more substantial for the heavier object than the lighter object, proportional to their mass difference. In other words, the user feels more energetic mechanical vibrations on their skin when interacting with a heavier object. We suspect users perceive these vibrations as a resistance force to acceleration (similar to the force of inertia), which leads them to perceive the mass of virtual objects.

The limitation of the force-controlled hand is that if we take two light virtual objects such that one object is twice as heavy as the other, it would be difficult to perceive the mass difference since both masses are well within the threshold of what the virtual hand can grasp and move around in the VR scene. However, with the presented vibration feedback, the vibration at the user's skin for the heavier object has twice the amplitude (Fig. 3). As a result, we expect that the user perceives the mass difference between the objects based on the vibration feedback.

## 5 EVALUATION

We evaluate our VR mass rendering techniques and verify our claims using both qualitative and quantitative measurements. We conducted a user study in which participants interact with virtual objects using the force-controlled co-located virtual hand and perform several object manipulation and comparison tasks. Moreover, we study the effect of the proposed vibration feedback on participants' ability to perceive virtual objects' masses and compare them based on the heaviness. More specifically, we look to assess these two hypotheses in our evaluations:

- Grasping and manipulating virtual objects using a co-located physically-based hand model in virtual reality gives a sense of mass perception and allows some degree of mass discrimination between virtual objects.

- The proposed vibration feedback can improve the sense of

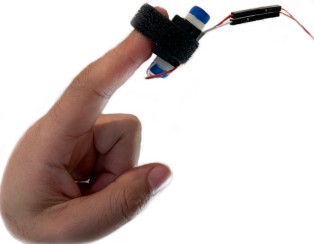

Figure 4: The voice coil actuator is strapped to the index fingertip of the user's dominant hand

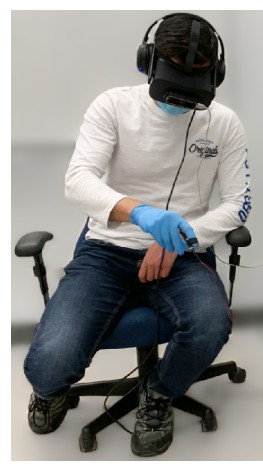

Figure 5: A participant interacting with a virtual object while wearing the VR headset with the Leap Motion hand tracker, VCA and nose-canceling headphones.

mass perception and enhance mass discrimination precision during virtual interactions between a physically-based virtual hand and virtual objects.

To examine the validity of the first hypothesis, participants perform virtual tasks involving interactions with objects with different mass values using the VR hand. However, evaluating these results of the VR hand interactions is not enough to validate our first hypothesis. The virtual environment runs in a physics engine, and users might get other clues to detect the difference in mass between objects that are not from the VR hand interactions only. These clues include: how the object interacts with each other, how they bounce when dropped on the virtual ground, and the speed at which they fall in the presence of air friction. To control the experiment for these additional cues, we ask participants to interact with each object individually and not push or touch an object using another. Additionally, we add a control interaction mode to our platform, called the spherical cursor. In this mode, instead of a co-located hand, users only see a spherical cursor co-located with the center of their palms. If the spherical cursor is within an object and the user puts their hand in a grasp pose, that object follows the cursor around the virtual scene until the user opens their hand. During grasping using the spherical cursor, we move the object by applying force to it in the cursor's direction. However, this force is proportional to the object's mass. As a result, objects with different mass follow the cursor at the same speed and acceleration. Therefore, comparing the quantitative and qualitative results from user interactions using a force-controlled hand versus the spherical cursor as a baseline allows us to validate the first hypothesis.

To test the second hypothesis, participants interact with virtual objects using the force-controlled hand both with and without the vibration feedback, which allows us to compare the results and analyze the effectiveness of the vibrotactile feedback in mass perception and discrimination.

### 5.1 Setup

In this subsection, we describe the study setup's hardware and software components and the range of mass values we use for our virtual objects. We use the MMXC-HF VCA by Tactile Labs, a relatively compact tactile actuator ($36mm \times 9.5mm \times 9.5mm$), and the Tactile Labs QuadAmp multi-channel signal amplifier. A pair of thin wires attached the VCA to the signal amplifier placed on a nearby table. The cables from the actuator point outwards from the user's finger, limiting the chance of cables touching the user's hands during virtual interactions. Using a 3d printed mount, we attach the voice coil actuator to the user's index fingertip (Fig. 4). We use the PC-powered Oculus Rift as our VR interface, which allows for external PC-based graphical computation. For tracking the user's hands, we attach a Leap Motion controller on the front side of the Oculus Rift VR headset for hand tracking.

In our system, we use the Bullet physics simulation [8] as our physics engine. One desirable feature of the Bullet library is that it permits the virtual hand's control by applying virtual force and torque from an external source. This feature enables us to implement the virtual coupling between our virtual hand and the tracked hand.

To render the virtual scene to the VR headset and work with the Bullet physics simulation, we use the Chai3D library. Chai3D [6] is a platform-agnostic haptics, visualization, and interactive real-time simulation library. Moreover, it supports visualizing using the Oculus Rift headset and has built-in Bullet physics integration, making it ideal for immersive and physically realistic haptic experiences.

In our study, we use cubes as our virtual object's shape since they are easier to grasp. During our experiments, there may be multiple virtual cubes in the scene with different mass ranges. For setting the mass range in our experiments, we should consider the physics engine that we use. The Bullet physics engine recommends keeping the mass of objects around 1 kg and avoid very large or small values [7]. Therefore, during our preliminary experiments, we set the virtual coupling coefficients so that users could pick up virtual cubes with masses up to 4 kg. However, past that mass point, it becomes too difficult to pick up the virtual cubes. Since we expect users to be able to interact and pick up any virtual cube in the scene, we chose 2.5 kg as our upper mass limit in our user studies for the heaviest objects and 0.25kg as our lower mass limit for the lightest objects.

### 5.2 Participants

Ten participants (5 female, 5 male) took part in this study. All participants were right-handed. Three participants had never used VR headsets before; one participant used them few times per week and the rest at most a few times per year. Seven of them had interacted with virtual objects during their VR experiences, and three had used haptic devices in VR games and applications. This study was approved by the University of Calgary Conjoint Faculties Research Ethics Board (REB18-0708). Participants received 20$ compensation for taking part in this user study.

### 5.3 Study

We begin the study by spending a few minutes ($<8$) familiarizing the participants with the VR headset, Leap Motion hand tracker, and the virtual study environment. After placing the haptic actuator on their dominant hand's fingertip, they practice how to pick up and move a virtual cube (1.25 kilograms) using the virtual co-located hand. We ask participants to always use their index fingers in grasping since the haptic actuator is attached to it. They are also encouraged to

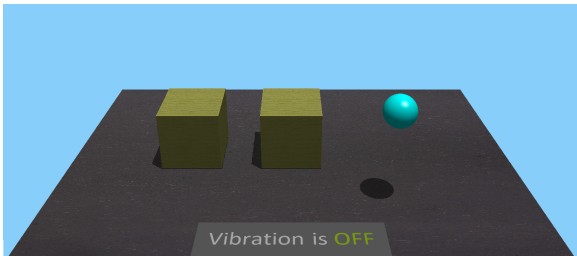

Figure 6: Two virtual cubes with random weights are placed in front of the participant to compare. The co-located spherical cursor mode is active, and the "Vibration Off" label indicates to the participant that they should not expect any vibration from the voice-coil actuator.

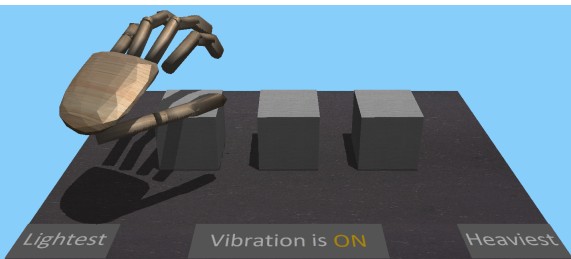

Figure 7: Three virtual cubes with random weights are placed in front of the participant to sort in ascending order from left to right. The "Vibration On" label indicates to the participant that they should expect vibration from the voice-coil actuator when picking up objects.

engage more fingers or tighten their grip to increase the grasping strength and move the training object around the scene both slowly and quickly. For consistency, we ask the participants only to use their dominant hand to interact with the virtual elements in the scene when the tasks start. During the virtual tasks, participants wear active noise-canceling headphones while white-noise is played through them to block any audible signal from the haptic actuator (Fig. 5).

In the first task, we present participants with six pairs of cubes and ask them to interact, grasp, move the objects, and think aloud about the experience. Furthermore, we ask them to compare the two cubes based on their mass and say if they feel they have the same mass or if one is slightly or considerably (or to whatever degree they perceive it) heavier than the other. Participants interact with virtual objects using the three interaction modes in the following order: spherical cursor, virtual hand without the vibration feedback, and virtual hand with the vibration feedback. As an example, Fig. 6 shows this task's setup while the interaction mode is set to the spherical cursor. For each interaction mode, participants compare two pairs of cubes. One pair has the largest mass difference given our mass range (0.25 and 2.5 kg), and the other pair has a smaller mass difference (0.25 and 0.5 kg). The system randomly decides if the smaller or larger mass difference pair is first presented to the user and randomly places the two cubes on the table for each set to avoid learning from the previous rounds.

In the next part, we ask participants to sort virtual cubes based on their mass. In sorting, a higher number of objects to sort means the participant spends more time picking up and moving objects around the scene, which results in a fuller user experience in comparing weights. However, a higher number of objects to sort increases the average time to complete the task, limiting the number of sorting rounds users can perform during a study session. Our preliminary experiments concluded that three cubes could offer a reasonable

balance between sorting time and user interaction with objects.

We quantized our mass range (0.25kg to 2.5kg) into two weight sets of size three. Having more than one weight-set allows a more in-depth analysis of the interaction modes across our mass range. Weber's law states that the difference in magnitude needed to discriminate between a base stimulus and other stimuli increases proportionally to the intensity of the base stimulus [12]. We can easily differentiate a 0.5kg mass versus a 1kg mass, but it is harder to distinguish a 10kg mass from a 10.5kg even though both pairs have the same weight difference. Therefore we chose our mass values with equal ratios between them using a geometric series. That gives a light weight-set (0.25kg, 0.44kg, 0.79kg) and a heavy weight-set (0.79kg, 1.4kg, 2.5kg).

Participants sort random permutations of the light and the heavy weight-set, using the three different interaction modes (spherical cursor, virtual hand without vibration feedback, the virtual hand with vibration feedback). Therefore we have six modes of sorting. As an example, Fig. 7 shows this task's setup while the interaction mode is set to the virtual hand with vibration feedback. In all sorting modes, three virtual cubes with similar appearance and size are placed on a virtual surface, and participants have to place them from left to right in ascending order based on the perceived mass. Participants perform six rounds of sorting for each mode. During each round, sorting modes are ordered randomly to remove the learning effect between the modes. Before the sorting task begins, we rotate between the modes to familiarize the participant with the scene. Furthermore, we ask participants to grasp each object at least once before finalizing their decision. Also, we recommend keeping each sorting under a minute; however, this is not a hard limit.

When the sorting task finishes, participants fill out a questionnaire regarding their experience during the two virtual tasks. After participants fill out the questionnaire, we ask them to elaborate on their answers during a semi-structured interview. Our post-session questionnaire is as follows: (each question is repeated for each of the interaction modes)

- While interacting with objects, I could perceive their mass. 1 to 5 (Strongly Disagree, Disagree, Neutral, Agree, Strongly Agree)
- I could feel one cube was heavier than the other. 1 to 5 (Strongly Disagree, Disagree, Neutral, Agree, Strongly Agree)
- How was your confidence level in sorting objects? 1 to 5 (Not confident at All, , , , Very Confident)
- How realistic were the interactions with objects? 1 to 5 (Very Unrealistic, Unrealistic, Neutral, Realistic, Very Realistic)
- Would you recommend experiencing the "" in VR games during interactions with virtual objects? 1 to 5 (Do Not Recommend at All, , Neutral, , Highly Recommend)

### 5.4 Results

We show the sorting results in the form of confusion matrices in Fig 8. Using the nonparametric Kruskal-Wallis test, we analyze the statistical significance of the difference between placement distributions of light, medium, and heavy objects for each of the sorting modes. For the spherical cursor (control mode), we observe statistically insignificant p-values of 0.463 for the heavy weight-set and 0.800 for the light weight-set, showing that the user could not discriminate between weights in this mode. For the virtual hand with no vibration feedback, we see statistically insignificant results for the light weight-set (p-value 0.928). However, for the heavy set, we see a significant effect of the virtual hand on sorting (p-value <0.001). In the case of sorting using the virtual hand with vibration feedback, we see a significant effect on sorting both for the light (p-value <0.001) and heavy (p-value <0.001) weight sets. To check the validation of the first hypothesis, we see a significant improvement for the heavy weight-set compared to the control mode (spherical cursor). However, the same cannot be said for the light

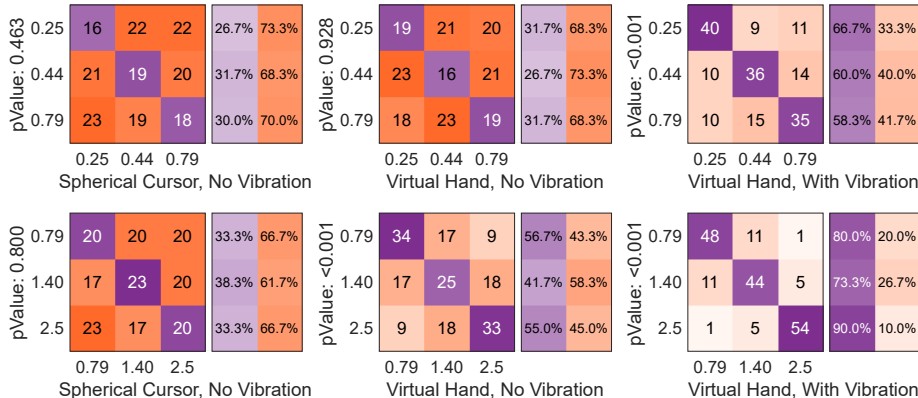

Figure 8: Sorting results of the six different sort modes in the form of confusion matrices. The top three matrices show the sorting results for the light weight-set (0.25kg, 0.44kg, 0.79kg), and the bottom three show the sorting results for the heavy weight-set (0.79kg, 1.4kg, 2.5kg). From left to right, matrices represent the three interaction modes (spherical cursor, virtual hand with no vibration, virtual hand with vibration). The matrices diagonals show the number of times the objects were sorted correctly.

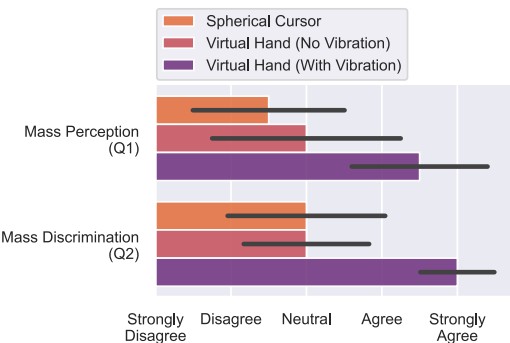

Figure 9: Users compare the sense of mass perception and discrimination between the three interaction modes in the post-session questionnaire. The bars represent the mean answer, and the black lines show the standard deviation.

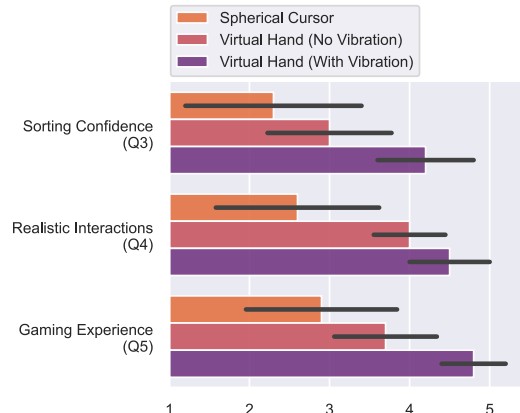

Figure 10: Users compare the sorting confidence, sense of realism, and gaming experience between the three interaction modes in the post-session questionnaire. The bars represent the mean answer, and the black lines show the standard deviation.

weight-set. To check for the second hypothesis, we see a statistically significant improvement in the light weight-set with the vibration feedback compared to only using the virtual hand. However, for the heavy set, we see significant effects both from virtual hand with and without the vibration feedback. Therefore, to check if the observed improvements in the precision of sorting for the light, medium and heavy objects are significant, we perform row by row comparison between the two confusion matrices using the Wilcoxon rank-sum test. Comparing the number of correct sorts for the heavy weight (54 correct sorts versus 33) gives a statistically significant p-value of $<0.001$, for the medium weight (44 correct sorts versus 25) p-value is $<0.001$, and for the light weight (48 correct sorts versus 34) p-value is $<0.01$, which shows that for the heavy weight-set the vibration feedback improvement is statistically significant as well.

The results of the questionnaire in Fig. 9 show that participants declared an improvement in mass perception and discrimination when the vibration feedback was enabled compare to only using the virtual hand. P6 (Participant #6) mentioned "With the hand no vibration, it was harder to tell the difference in mass, but I think you could still, it was realistic enough that it was engaging, but the vibration one I'm not if it's like a mental thing, it just helps a lot more with the differentiating between the different masses and the movements". We also see neutral results for the spherical cursor. Generally, participants mentioned they could not differentiate between the objects using the spherical cursor. P2 mentioned, "It was

harder for me to use the cursor to compare the weights, most of the time I thought they were like identical". For the virtual hand without the vibration feedback, participants on average expressed neutral opinions regarding its ability to give them the sense of mass perception and discrimination. However, the results from the sorting task show they performed better than the control. Also, some participants mentioned different encounters that enabled them to differentiate between weights. P5 mentioned "I'm picking it up, how long would it slide, ok hold it, I shake it around it slides faster ... if I hold it, it slips faster then it's heavier", and P6 said "(with the virtual hand) if I grab it loose the heavy one just drops as opposed to the light one stays in even if I'm shaking it", and "looking at the movement, if I'm moving my hand it's a bit slower it just feels heavier versus if it's a quick it just feels lighter"

Fig. 10 shows that participants expressed having more confidence in sorting when the vibration feedback was enabled. However, without the vibration feedback, they expressed neutral confidence. Furthermore, participants generally stated that the vibration feedback added to the interaction's realism and that the virtual hand's interactions were realistic. P4 said "For the vibration also, I felt like it helped me, felt like it's more real, I'm touching things, not just I'm seeing that I'm touching things". Furthermore, participants ex-

pressed interest in experiencing the vibration effect in virtual reality games.

Finally, we asked the participants how did interaction with virtual objects feel when they vibrated. P2 said: "if felt like it has resistancy to move, based on that I felt like it's heavier, might be heavier" and P7 mentioned "When I picked a cube with vibration, I could feel that something is trying to, I don't know, annoy me bother me, might be something like the gravity taking it back to the ground, it feels that I should put more energy to pick it up" and further elaborated "the one that without vibration I just pick it with two fingers I played with that, but the one with vibration when I tried to pick it with two fingers, suddenly I tried to keep it with all my fingers because I thought that it might slides and drops."

Overall our findings indicate that the presence of the force-controlled virtual hand both with and without the vibration effect gives a sense of weight discrimination and perception. However, the virtual hand without vibration feedback is only effective for heavier objects closer to the hand strength threshold. Furthermore, the virtual hand with the vibration effect improves the weight perception and discrimination sense for both lighter and heavier objects without having a negative effect on the realism of the experience. Therefore, our results validate our hypotheses.

## 6 CONCLUSION

Rendering the mass of objects in virtual reality without limiting the hand movements is a challenging task. In this paper, we propose using a force-controlled hand in VR to give a sense of mass perception and discrimination by enabling physically realistic hand-object interactions. We also propose a complementary vibration effect proportional to the object's mass and acceleration to improve the sense of mass perception and discrimination. We conducted a user study and performed qualitative and quantitative analysis, which indicates that our hypotheses are valid. The physically-based virtual hand can give a sense of mass perception and discrimination for heavier objects closer to the upper limit of its grasping strength. Furthermore, the vibration feedback greatly enhances the mass perception and discrimination for a wider mass range in our study while improving the interaction's realism.

## 7 FUTURE WORKS

One potential future direction for this research is to analyze the mass discrimination ability for the virtual hand and the vibration effect for a broader mass range and different mass ratios between the objects. Moreover, we are interested in analyzing the vibration effect's behavioral effects on the user's movements during virtual interactions.

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
