# OpenReview forum: "Simulating Mass in Virtual Reality using Physically-Based Hand-Object Interactions with Vibration Feedback"
_graphicsinterface.org/Graphics_Interface/2021/Conference/Second_Cycle — GI 2021_

### Official Review · Reviewer_nRss · 2021-04-26
**Evaluating of using simulating mass and vibration to create haptic feedback**

**Rating:** 7
**Confidence:** 3

**Review:**

The paper is about a physically-based virtual hand and a vibrotactile actuator to emulate the sensation of mass on a virtual object. The paper has two parts. The first part describes how the physical-based virtual hand works and justifies the inclusion of the vibrotactile actuator in the interface. The second part is an evaluation of this interface and how it affects the mass perception and discrimination of virtual objects. Results show that using the physical-based virtual hand was better than the base condition and that using the vibrotactile actuator allows users to discriminate both heavy and light objects.

In general, the paper is well written and easy to follow. I am not an expert in the area, so I cannot talk about the device novelty, but the description is detailed enough that someone else could implement this technique. Yet, I encourage the authors to include hand diagrams specifying the direction of the force, which will make the description easier to understand, even for people outside the area.

My main concern with the paper is about missing information from the user study. First, the number of participants is low (only 10). Based on this, I ask the authors to include the effect size of the results and the statistical power of the user study. Another question I have is about the base condition. Using a spherical cursor does not seem to be a standard base case for evaluating mass emulation. Therefore, I encourage the authors to include a description of the possible variables that could affect the results in the discussion, e.g., colour, shape, size, position in the hand, etc. My third question is about the minor mass limit and how it might have affected the results. According to the paper, the engine recommends keeping the mass of objects around 1 kg. Yet, in the light-weight-set, the mass of all the objects is under 1 kg. Finally, I encourage the authors to include references to other papers in the discussion section to help place their results in context to previous work.

Besides these problems, the paper results are interesting, and the paper will be a good addition to the conference.

---

### Official Review · Reviewer_Kzh4 · 2021-04-30
**This is a really well put together study which looks at the feasibility of simulating mass for physical hand-object interaction.**

**Rating:** 9
**Confidence:** 4

**Review:**

This is a really well written, novel study, well structured and presented. The study design is good, although ideally more than 10 participants would have been nice, nevertheless a really well designed study and well deployed. Overall, the background and related work section covers a good range of contemporary content and sets the study up well within the context of the state of the art. A really good and thorough formal presentation of the system is provided and a really comprehensive evaluation is also carried out, although participant numbers could have been higher, and perhaps some additional qualitative interview data could have complemented the questionnaire data that was collected, the study is still really well formulated and explores a really interesting and novel area. I believe the results are of significant value and should be published.

---

### Official Review · Reviewer_a3XE · 2021-05-03
**A nice evaluation of visual and haptic cues**

**Rating:** 7
**Confidence:** 3

**Review:**

In this paper, the authors propose using physics-based cues (acceleration ranges linked to weight) with vibrotactile cues to enable users to accurately detect relative object weights. They clearly describe their two techniques and conduct a study that shows that visual cues aid in discrimination of heavier weights, but not lighter, and that haptics (via vibrotactile feedback) allows users to more accurately discriminate heavier objects and to distinguished medium/lighter objects.

I'm not 100% familiar with all of the research in this area on pseudophysics and haptics to communicate varying weight ranges to users within virtual environments. However, assuming that the techniques presented by the authors are novel (or their combination is novel), then I believe that this paper merits acceptance.

The overall study design was good with, essentially, a training phase followed by six rounds of a sorting phase where the six different sorting tasks (three techniques by two weight levels) were randomly ordered. The confusion matrices for the sorting results of these randomly ordered tasks show a clear effect. Overall, there is little to say that is negative about this work provided its novelty is good: the ideas seem sound, the combination of the visual and haptic effects seems novel, the study is well designed and controls confounds to a level sufficient to give me confidence the effects are real, and the overall technique holds promise for increased richness within virtual environments via an ability to more accurately discriminate relative weights of objects.

---

### Meta-Review · Area_Chair_E45J · 2021-05-06

**Recommendation:** Accept
**Confidence:** 4

**Metareview:**

All reviewers agree that the paper’s quality is high. For example, R1 found the study design well done. R2 also found the paper well written and well structured. And R3 state that the description of the device is good.

However, the reviewers asked for additional information to make the paper even stronger. Here is a list below

Number of participants, and effect size of the results – R2, R3
Add diagrams of the force directions – R3
Add additional information about the user study – R3
Incorporate more previous work into the discussion – R3

I encourage the authors to review their comments and add the request information if possible.

---

### Decision · Program_Chairs · 2021-05-08

Accept